# Clocked dynamics in artificial spin ice

Johannes H. Jensen [1,4] ✉, Anders Strømberg [2,4] ✉, Ida Breivik [2], Arthur Penty [1], Miguel Angel Niño [3], Muhammad Waqas Khaliq[3], Michael Foerster [3], Gunnar Tufte [1] & Erik Folven [2]

Artificial spin ice (ASI) are nanomagnetic metamaterials with a wide range of emergent properties. Through local interactions, the magnetization of the nanomagnets self-organize into extended magnetic domains. However, controlling when, where and how domains change has proven difficult, yet is crucial for technological applications. Here, we introduce astroid clocking, which offers significant control of ASI dynamics in both time and space. Astroid clocking unlocks a discrete, step-wise and gradual dynamical process within the metamaterial. Notably, our method employs global fields to selectively manipulate local features within the ASI. Sequences of these clock fields drive domain dynamics. We demonstrate, experimentally and in simulations, how astroid clocking of pinwheel ASI enables ferromagnetic domains to be gradually grown or reversed at will. Richer dynamics arise when the clock protocol allows both growth and reversal to occur simultaneously. With astroid clocking, complex spatio-temporal behaviors of magnetic metamaterials become easily controllable with high fidelity.

Artificial spin ice (ASI) are systems of coupled nanomagnets arranged on a two-dimensional lattice. The nanomagnets are elongated, giving them two stable magnetization directions, thus behaving as artificial spins. Dipolar interactions give rise to a rich variety of emergent behavior, as determined by the ASI geometry[1–3], such as domains of long-range order. As this behavior can be probed directly, ASIs have attracted considerable interest as model systems for the study of fundamental physics[4,5]. More recently, ASIs have shown promise for device applications, such as substrates for computation[6–12].

Controlling when, where and how ASIs change state is instrumental to both fundamental research and applications. Control of the state evolution would enable an experimenter to access a richer variety of emergent ASI phenomena. Furthermore, such control will be key in applications such as neuromorphic computing, where the functionality is derived directly from the evolving magnetic state of the ASI. However, external control of emergent ASI dynamics has so far proven difficult.

External fields are the primary method used to perturb ASIs in a controlled manner. Various global field protocols have been employed.

For example, a cycled in-plane field is often used to characterize magnetization reversal[1,13–21]. Another approach is to use a rotating field with slowly decreasing amplitude to effectively anneal the ASI to a low energy state[22–29]. While there are variations of these simple field protocols, more complex protocols are largely unexplored.

These approaches use field strength to modulate ASI behavior, which will typically result in uncontrolled avalanches of activity[30]. An in-plane field will advance ASI state primarily when the strength of the field is increased beyond the coercivity of the array, and is highly dependent on field resolution[19,25,29]. Consequently, the discrete spin flip dynamics in the ASI are sudden and hard to control.

In this work, we introduce a field protocol scheme called astroid clocking, which produces fundamentally different spin flip dynamics. In contrast to previous approaches, astroid clocking unlocks a discrete, step-wise and gradual evolution of spin states. The method offers significant control and understanding of the dynamical process in both time and space. Key to the method is exploiting the shape and orientation of the nanomagnet switching astroids, together with their dipolar coupling. Specific field angles can then be established to

[1]Department of Computer Science, Norwegian University of Science and Technology, Trondheim, Norway. [2]Department of Electronic Systems, Norwegian University of Science and Technology, Trondheim, Norway. [3]ALBA Synchrotron Light Facility, Carrer de la Llum 2 – 26, Cerdanyola del Vallés 08290 Barcelona, Spain. [4]These authors contributed equally: Johannes H. Jensen, Anders Strømberg. ✉e-mail: johannes.jensen@ntnu.no; anders.stromberg@ntnu.no

selectively address emergent local features within the ensemble, such as domain boundaries. A clock protocol which pulses fields in an alternate fashion at these angles, is then used for driving the intrinsic dynamics of the ASI. Distinctively, the clock pulses maintain a constant field amplitude.

In the context of nanomagnetic logic, Nomura et al.[31] demonstrated how the shape of two overlapping Stoner-Wohlfarth astroids can be exploited to preferentially switch nanomagnets in a 1D shift register. Astroid clocking extends and generalizes this concept to 2D nanomagnet arrays, and non-elliptical nanomagnets with different astroid shapes. We show how astroid clocking reveals the intrinsic dynamics of coupled nanomagnetic systems.

In this study we consider the pinwheel ASI system, but stress that astroid clocking is readily applicable to other coupled nanomagnetic systems as well. We have, for instance, obtained promising results with both square and kagome ASI in simulations. Here, we demonstrate and analyse how ferromagnetic domains in pinwheel ASI can be gradually grown and reversed at will using astroid clocking. Different clock protocols are explored, giving rise to distinct properties of the spin flip dynamics.

## Results

### Astroid clocking

Pinwheel ASI[2,19,32] consists of nanomagnets arranged on two interleaved square sublattices, as shown in Fig. 1a. In this study, the magnets in the two sublattices $L_a$ and $L_b$ are rotated +45° and −45° with respect to the array edges. The sublattice and magnetization of the magnets are indicated by their color: magnets in sublattice $L_a$ are orange or blue, while magnets in sublattice $L_b$ are pink or green. For brevity, we will refer to magnet state by these four colors.

Pinwheel ASI favors a ferromagnetic ordering, with emergent domains of coherent magnetization. Figure 1a shows the four possible domain directions: rightwards (orange/pink), leftwards (blue/green) and so on. The ferromagnetic domains are separated by domain walls, which are slightly less energetically favorable[32].

The switching threshold of a nanomagnet depends on the field angle, and can be approximated by the Stoner-Wohlfarth astroid. Figure 1b shows the switching astroids for the two orientations of stadium-shaped magnets in pinwheel ASI[33]. A magnet will switch state if the total field acting on it lies outside the astroid boundary, and the field is directed against the current magnetization. Note that we use the term *switching astroid* to refer to any angle-dependent threshold curve, also when its shape deviates significantly from the ideal geometric astroid shape. The compound shape in Fig. 1b resembles an ideal astroid, but is in fact two overlapping, highly distorted astroids (solid and dashed outlines).

Nanomagnet shape largely determines the shape of the astroid. Stadium-shaped nanomagnets, commonly used in ASI, have a switching astroid with 2-fold rotational symmetry[33]. This is in contrast to classical Stoner-Wohlfarth astroids that display 4-fold rotational symmetry derived for elliptical nanomagnets[34].

Switching astroids that break the 4-fold rotational symmetry, can be exploited to selectively address nanomagnets that are rotated 90° with respect to each other. If the total field lies within the shaded regions in Fig. 1b, only the nanomagnets in the corresponding sublattice will be able to switch. A field in the orange/blue shaded regions will address only the magnets in sublattice $L_a$, while a field in the pink/green regions will address only magnets in $L_b$. Furthermore, each region promotes a specific magnet state within each sublattice, e.g., a field in the blue shaded region promotes blue magnets by switching orange magnets.

In this study, we define two *bipolar clocks* A and B along the +22° and −22° axes, respectively. As shown in Fig. 1b, each clock consists of a positive and negative *clock field* of magnitude H along the clock axis. The four arrows in Fig. 1b are colored according to the magnet states they promote, e.g., the $\mathbf{H}_A$ field only promotes orange magnets.

The dipolar fields $\mathbf{h}_{dip}$ from neighboring magnets may either promote or prevent switching. If the dipolar fields are directed out of (into) the astroid, they effectively promote (prevent) switching. A clock field can thus selectively address a subset of a sublattice,

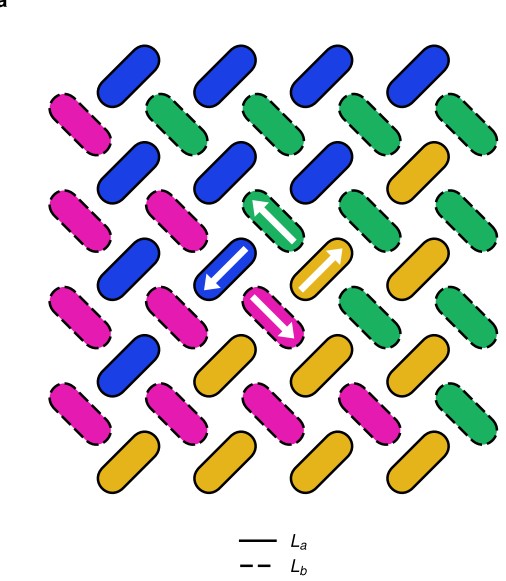

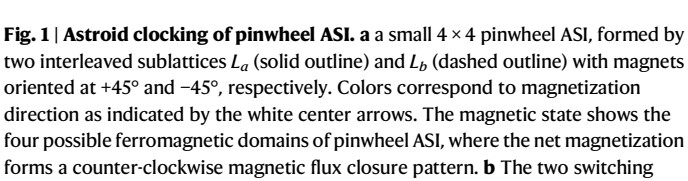

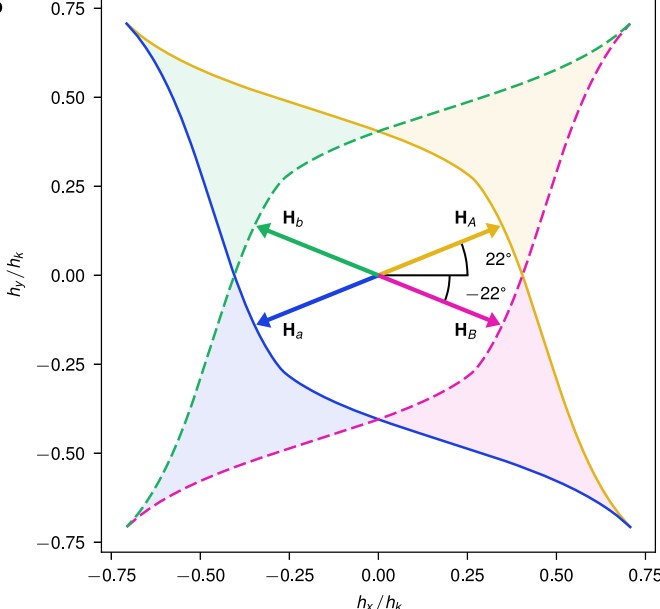

**Fig. 1 | Astroid clocking of pinwheel ASI. a** a small 4 × 4 pinwheel ASI, formed by two interleaved sublattices $L_a$ (solid outline) and $L_b$ (dashed outline) with magnets oriented at +45° and −45°, respectively. Colors correspond to magnetization direction as indicated by the white center arrows. The magnetic state shows the four possible ferromagnetic domains of pinwheel ASI, where the net magnetization forms a counter-clockwise magnetic flux closure pattern. **b** The two switching astroids for the magnets in sublattice $L_a$ (solid lines) and $L_b$ (dashed lines), along with the four clock fields, $\mathbf{H}_A$, $\mathbf{H}_B$, $\mathbf{H}_a$, and $\mathbf{H}_b$. The astroid edges are colored according to the magnet state which is promoted when fields cross the edge. Similarly, the colored regions correspond to fields that exclusively promote a magnet state within a sublattice. Astroid axes are normalized with respect to the hard axis switching threshold, $h_k$.

depending on the state of the ensemble. The clock angles ±22° are selected to allow the dipolar fields to have a large influence on switching, using a field strength $H$ close to the switching threshold. However, a precise angle is not crucial and the method tolerates a wide range of clock angles. Our system tolerates clock angles in the range 10° to 35°, and field strengths accurate to within 3 mT to 4 mT. Other clock angles, such as ±45°, would appear to allow for similar switching selectivity. However, as we shall see later, the influence of dipolar fields renders these angles unsuitable.

Figure 2 illustrates astroid clocking, where a *clock pulse* is defined as ramping a clock field from zero to $H$ and down to zero again. The ramping speed is much slower than the timescale of nanomagnetic switching. A *clock protocol* is a specific sequence of clock pulses. For example, *AB* clocking consists of repeated alternating clock pulses of *A* and *B*. We define a *clock cycle* as a single sequence of the clock pulses in a protocol, e.g., an *aAbB* clock cycle is the sequence of four pulses (*a*, *A*,

*b*, *B*). A *unipolar* clock protocol exclusively employs one polarity of each clock, while a *bipolar* clock protocol employs both polarities.

## Unipolar clocking

First, we explore the spin flip dynamics of pinwheel ASI when subject to the unipolar clock protocols *AB* and *ab*. The 50 × 50 pinwheel ASI (5100 magnets) is initialized with a small rightwards (orange/pink) domain in the center of an otherwise leftwards polarized (blue/green) array. Figure 3 (1) shows a closeup of the initial state.

Figure 3 (2–8) shows the state evolution of the array subject to *AB* clocking, obtained from flatspin simulations (see Methods). As expected, the *A* pulse selectively switches magnets in sublattice $L_a$ from blue to orange, while the *B* pulse selectively switches magnets in sublattice $L_b$ from green to pink.

Interestingly, the particular magnets that switch are the ones along the domain wall. As a result, the inner (leftwards) domain grows

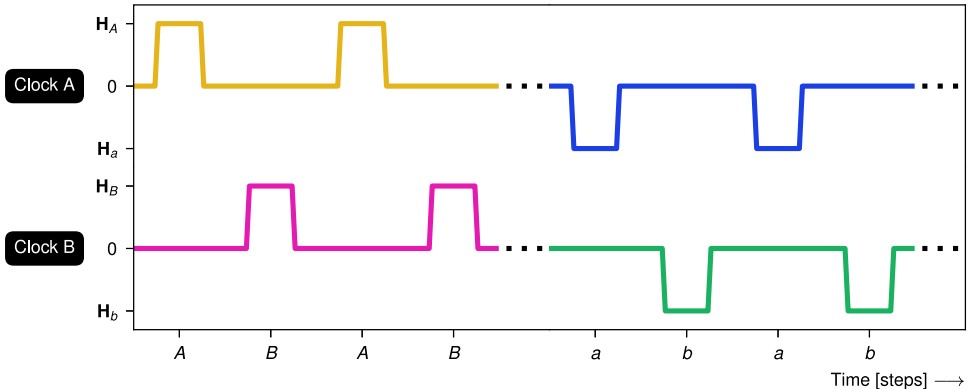

**Fig. 2 | Clock diagram of astroid clocking.** Clock protocols are defined by sequences of clock pulses. The clock diagram shows *AB* clocking (alternating pulses of the positive clock fields $\mathbf{H}_A$ and $\mathbf{H}_B$) followed by *ab* clocking (alternating pulses of the negative clock fields $\mathbf{H}_a$ and $\mathbf{H}_b$).

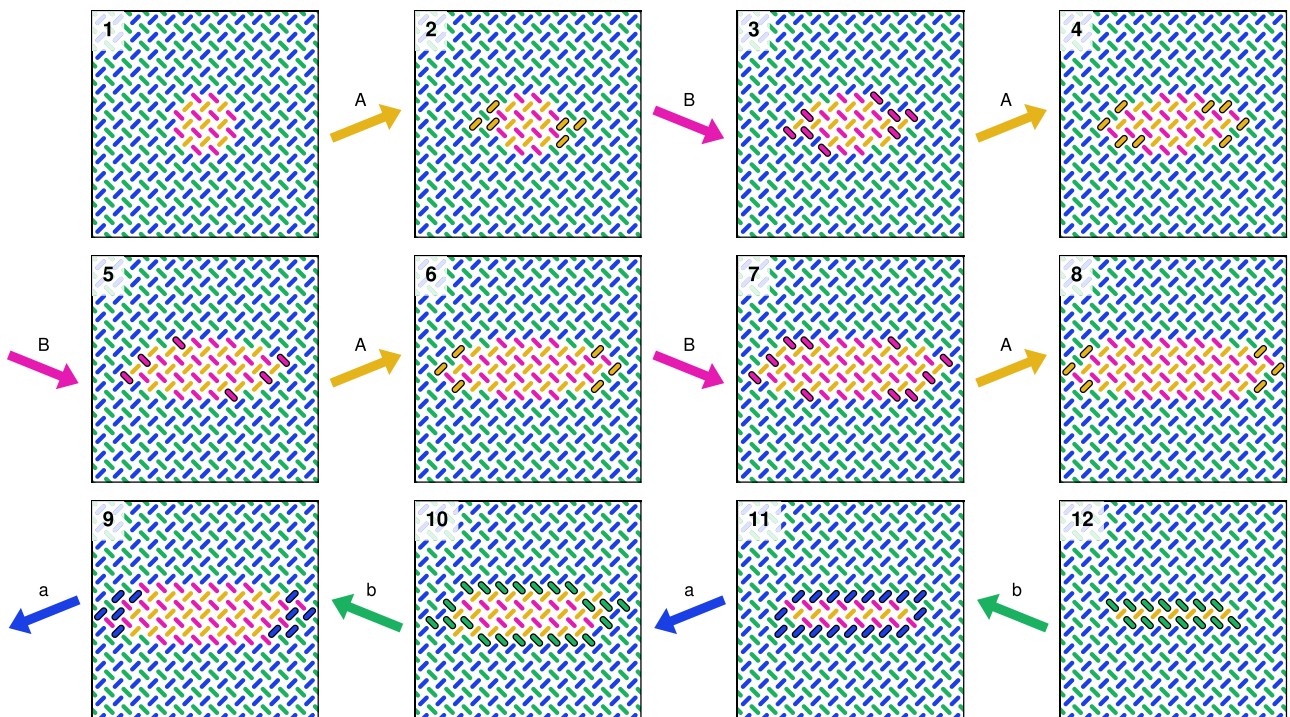

**Fig. 3 | Simulation of unipolar astroid clocking of pinwheel ASI in flatspin.** Each snapshot shows a zoomed-in view of the 50 × 50 nanomagnet system, at different points during a clock protocol. (1) shows the initial state, a small orange/pink (rightwards) domain in the center of an otherwise polarized blue/green (leftwards) array. (2–8) show the state during *AB* clocking, resulting in gradual domain growth. (9–12) show the subsequent states during *ab* clocking, resulting in gradual domain reversal. Magnets that change state between snapshots are highlighted with a solid black outline.

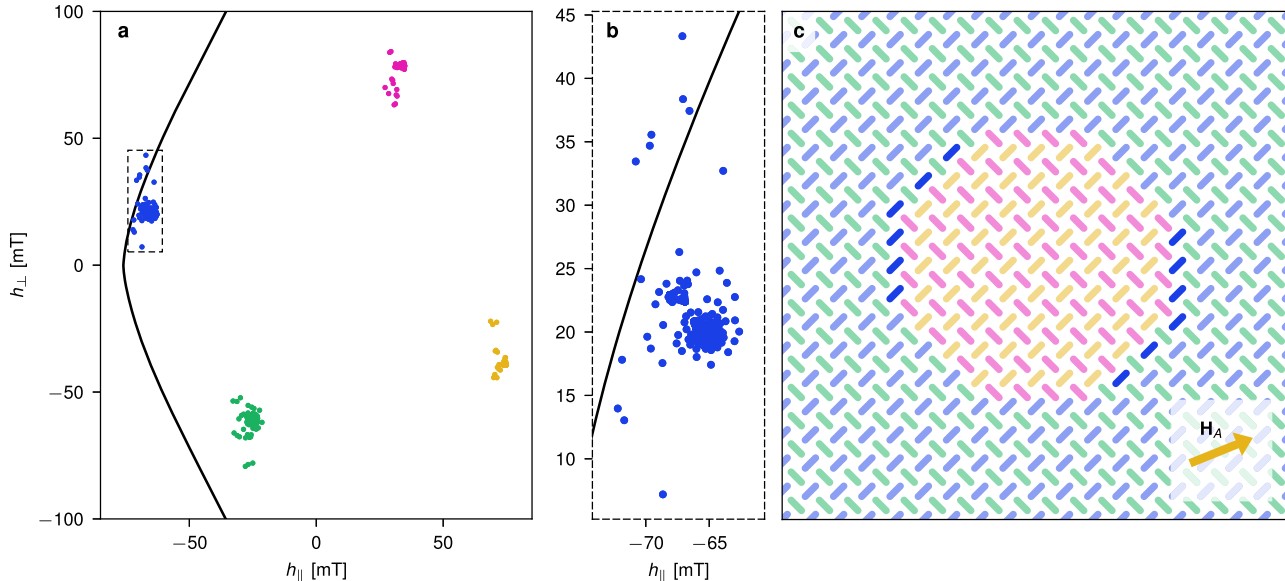

**Fig. 4 | Astroid clusters showing relative locations of all the magnets within their respective switching astroids.** The plots show clusters and astroid (black curve) for the pinwheel system shown in (**c**), when subject to the clock field $\mathbf{H}_A$. **a** astroid cluster plot where each dot represents the total field $\mathbf{h}_i = \mathbf{H}_A + \mathbf{h}_{dip}^{(i)}$ experienced by a magnet $i$, projected onto its parallel ($h_\parallel$) and perpendicular ($h_\perp$) axis. Note that the positive direction of the parallel component is with respect to the magnetization direction of each nanomagnet. The colors in the plot correspond to magnet state. Note that the astroid plot shows location relative to each magnet's own switching threshold, e.g., orange magnets are far from switching as they are aligned with $\mathbf{H}_A$. **b** closeup of the blue cluster, revealing a sub-group of blue magnets that lie outside the switching astroid and are eligible for switching. These magnets are highlighted in **c**, and are all found to lie along the vertical and +45° domain walls. The data is obtained from flatspin simulations.

gradually over time, with only a thin layer of the domain advancing after each clock pulse. The growth is monotonic and step-wise, driven by the clock pulses.

A curious property is that the domain grows mainly in the horizontal direction. In Fig. 3 (2–8), the magnets along the vertical domain walls are the only ones to switch. If the growing domain reaches the edges of the array, the direction of growth changes and becomes vertical, eventually filling the entire array (Fig. 5).

Inverting the clock pulses (*ab* clocking), will instead grow the outer (blue/green) domain and consequently shrink (reverse) the inner (orange/pink) domain. As can be seen in Fig. 3 (9–12), domain reversal from (8) proceeds in both vertical and horizontal directions, resulting in reversal of the inner domain in fewer clock cycles compared to growth. Hence there is an apparent asymmetry in the direction of domain growth and reversal.

**Growth and reversal mechanism**

To understand the mechanism behind the domain growth and reversal, we consider the larger domain shown in Fig. 4c, subject to $\mathbf{H}_A$. In Fig. 4a, we plot the relative locations of all the magnets within their respective switching astroids. Each dot represents the total field $\mathbf{h}_i = \mathbf{H}_A + \mathbf{h}_{dip}^{(i)}$ experienced by a magnet $i$ in its local frame of reference. There are four clusters of dots within the astroids, corresponding to the four magnet colors, where only the blue magnets are close to switching.

The internal structure of each astroid cluster is a result of the nanomagnet dipolar coupling, and a direct consequence of the ASI geometry. In the absence of dipolar fields, each cluster collapses into a single point. The dipolar fields add complex structure to the clusters, with sub-groups corresponding to different subsets of magnets within the ASI. For a detailed analysis of neighbor contributions, see Supplementary Discussion.

The inset shown in Fig. 4b reveals the structure of the blue cluster. Notice there are a few blue dots that lie outside the astroid, corresponding to magnets that are eligible for switching, which are

highlighted in Fig. 4c. Evidently, the switchable magnets all lie along the vertical and +45° domain walls.

When a magnet switches, its location within the astroid jumps to the cluster of opposite spin, e.g., a blue magnet switches to the orange state. In addition, neighboring magnets will see a change in the dipolar fields, causing movement within their respective clusters. In this way, the switching of a magnet may enable future switching in neighboring magnets, either during the current or a future clock pulse. As can be seen in Fig. 4b, the mechanism hinges on both the internal structure of the astroid clusters, and at what angle the cluster approaches the astroid edge. Clock angles around 22° work because they allow only magnets along certain domain walls to switch. Other clock angles, such as ±45°, are unsuitable since large parts of the astroid cluster will lie close to the astroid edge (at $h_\perp = 0$), resulting in avalanches of switching within a sublattice.

The observed horizontal domain growth can now be explained from the internal structure of the astroid clusters. We have seen that magnets along certain types of domain walls can be selectively switched under an applied clock field. Switching the blue (highlighted in Fig. 4c) magnets along these domain walls reverses their dipolar fields, which affects the structure of the green cluster. Consequently, green magnets that are part of the domain walls will approach the switching astroid. When the $\mathbf{H}_B$ field is subsequently applied, these magnets will be outside the astroid and hence switch. As this cycle repeats, the result is an apparent horizontal domain growth emerging from the dipolar interactions and clock fields.

During domain reversal, both horizontal and vertical domain walls take part in the process. As a result, reversal requires fewer clock cycles compared to growth. During reversal, the switchable magnets lie along both the horizontal, vertical and −45° domain walls (Supplementary Fig. 1).

We find that the horizontal domain wall movement, particular to reversal, is dependent on the curvature of the reversing domain. If the horizontal domain wall is surrounded by a blue/green domain on three sides, there is a stronger dipolar "push" towards the astroid

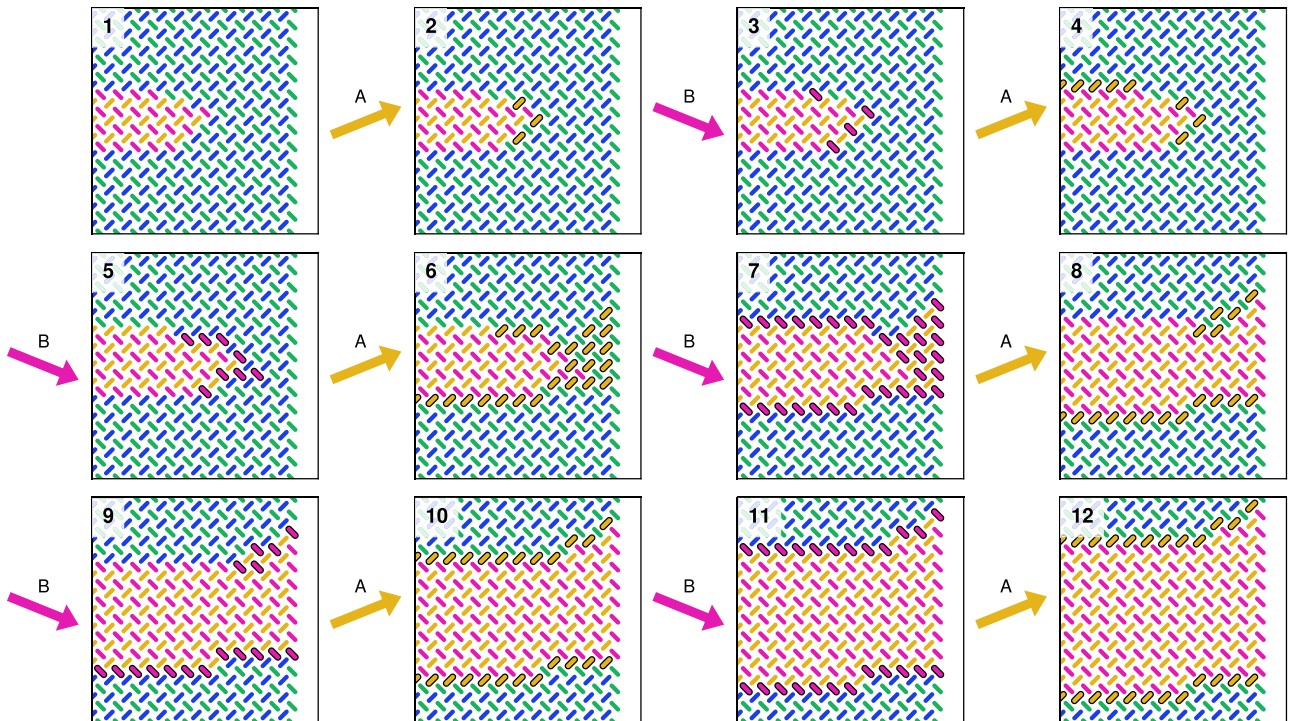

**Fig. 5 | Unipolar *AB* clocking of an orange/pink (rightwards) domain as it reaches the edge of the array.** There is an apparent transition from horizontal to vertical domain growth (5–6). Vertical growth proceeds by avalanches of spin flips, starting at the bottom-left and top-right corners of the domain at the array edge. Magnets that change state between snapshots are highlighted by a solid black outline.

edge. As such, domain shape plays a crucial role in the reversal process.

When a domain grows to reach the edges of the array, there is an apparent transition from horizontal to vertical growth (Fig. 5). We find that vertical growth proceeds by avalanches along the domain wall, starting at the bottom-left and top-right corners of the domain, close to the array edges.

## Experimental growth and reversal

Next, we demonstrate astroid clocking of pinwheel ASI experimentally. Samples are imaged with x-ray magnetic circular dichroism photoemission electron microscopy (XMCD-PEEM), with an in-situ vector magnet to perform astroid clocking. See Methods for details.

After polarizing all magnets in the leftwards direction (bright contrast), we perform steps of *AB* clocking, imaging in-between each clock cycle. Figure 6a shows total magnetization of the ensemble, obtained from the XMCD-PEEM images, which increases in a stable, monotonic fashion. Selected experiment snapshots are shown in Fig. 6b. Snapshots (1–3) show that domains nucleate at the vertical edges then predominantly expand horizontally.

Domain formation at the vertical array edges can be explained by the dipolar field-driven mechanism behind *AB* clocking. While domain nucleation along horizontal edges is possible, continued growth primarily occurs in the horizontal direction, preventing further expansion of horizontal edge nucleated domains. Spatial control of nucleation could be achieved, e.g., by changing the edge geometry, introducing internal edges or reducing the coercivity of selected magnets.

In any physical ASI system, the nanomagnets will exhibit a range of intrinsic switching thresholds, a *disorder*, due to imperfections and microscopic variations of material composition. Disorder affects both domain shape and growth dynamics, as evident in our experimental results. Compared to the idealized simulations, domains appear more organic, with distinct features such as jagged edges, slanted domain walls, and sporadic holes. In terms of dynamics, some domain borders

get stuck for several clock cycles, while others advance more than one step during a single cycle (see Fig. 7).

By introducing disorder to the simulations (see Methods), we obtain results that more closely resemble the experiment. The magnetization curve and snapshots from simulations with disorder are included in Fig. 6. Notice how the simulated snapshots show organic-looking domains that resemble the domains of the experiment.

After growth, we apply the reversal clock protocol, *ab* clocking. For each *ab* clock cycle, the magnetization reduces sharply, with domains shrinking more rapidly compared to the increase during growth. Comparing snapshot (3) and (4) of Fig. 6b, it is clear that the domains shrink in both vertical and horizontal directions.

Next, we conduct a control experiment to verify that simply repeating a clock pulse *A* or *B* does not result in domain growth. After re-initializing the system, we apply several pulses of *A*, then several pulses of *B*, imaging after each pulse. As seen in the last part of Fig. 6a, only the first application of *A* or *B* results in growth (see also Supplementary Fig. 2). Growth progresses only when the type of clock pulse is changed, which confirms that the alternating pattern of *A* and *B* is what drives the observed domain growth.

These experiments affirm the viability of astroid clocking in the face of experimental sensitivities (as low as < 1 mT from Fig. 4) and potential impediments such as fabrication imperfections, temperature effects, and material degradation. While unstable individual magnets and inaccuracies in the image analysis induce some noise, it is negligible compared to the effect of astroid clocking. Experimental astroid clocking is surprisingly robust, demonstrating that it is possible to precisely control the spin flip dynamics of ASIs using global fields.

## Bipolar clocking

In bipolar clocking, each clock may be pulsed in both polarities. We consider two clock protocols illustrated in Fig. 8, namely *aAbB* and its inverse, *AaBb* clocking. In contrast to unipolar clocking, the magnetic fields in these bipolar clock protocols are balanced, i.e., the sum of all

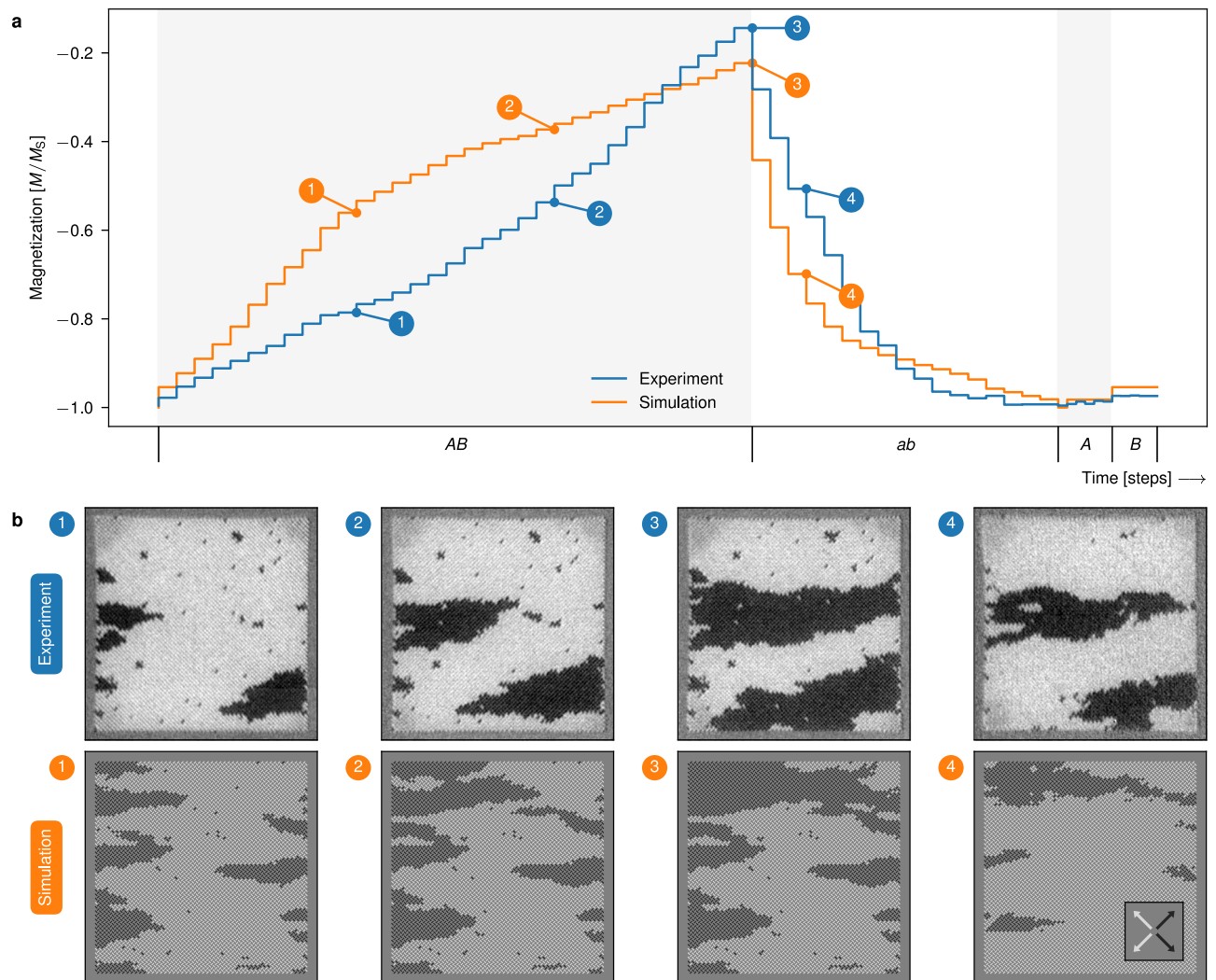

**Fig. 6 | Results of growth and reversal with unipolar clock protocols, and control experiment. a** total magnetization of the ensembles subject to the different clock protocols. The timeline indicates clock time, labeled by the clock protocol. During *AB* clocking, the ensembles undergo growth and hence an increase in magnetization. The second phase, *ab* clocking, quickly reverses domains and total magnetization. The control experiment, consisting of separate *A* and *B* clock sequences, show no development of the domains. **b** magnetic image snapshots (experiemenical XMCD-PEEM images and flatspin simulated XMCD-PEEM contrast images) of the ensembles at the specified points in time. The depicted ensembles are approximately 12.5 μm × 12.5 μm (50 × 50 pinwheel ASI, 5100 magnets). All XMCD-PEEM images are available in Fig. 7 and Supplementary Fig. 2. Videos of the experiment and simulation are provided in Supplementary Movies 1 and 2.

clock fields is zero. One might then expect that this results in a net zero magnetization change.

On the contrary, bipolar clocking also results in domain growth and reversal, and a net change in magnetization. Figure 9a plots the total magnetization of pinwheel ASI subject to bipolar clocking. As can be seen, *aAbB* clocking results in net domain growth, while *AaBb* clocking results in domain reversal.

In contrast to unipolar clocking, bipolar clocking can also induce morphological changes to the growing domains. As a result of the bipolarity of the clock pulses, domains are now able to both grow and shrink within the same clock cycle. In the experiment snapshots of Fig. 9b, we observe growth from (1) to (2), followed by a clear change in domain morphology from (2) to (3), and further growth between (3) to (4). In simulations, we can observe the step-wise details of simultaneous growth and morphology changes, as shown in the zoomed in snapshots. Inverting the clock protocol (*AaBb* clocking) results in domain reversal.

The deciding factor for growth or reversal is the polarity of the last clock pulse at the transition between the two clocks. Each clock in

*aAbB* clocking, for example, ends on the positive polarity at the transition (*aA* and *bB*), resulting in growth of the rightwards (orange/pink) domains.

Within a bipolar clock cycle, there is an apparent competition between growth and reversal. Some domain wall configurations result in net domain growth (others in net reversal), in a "one step back, two steps forward" process (see Supplementary Discussion). In this way, a domain may grow horizontally and reverse vertically, thereby gradually changing shape over time (see Fig. 10). While the balance between growth and reversal can be delicate, there is a clear trend for the clock protocols explored here, namely growth for *aAbB*, and reversal for *AaBb*.

Compared to unipolar clocking, the dynamics in bipolar clocked pinwheel ASIs are more varied and complex. While there is a gradual net domain growth, the activity can intermittently spike and linger, depending on the particular state of the ensemble (see Supplementary Movies 3 and 4). Bipolar clocking hence unlocks a wide variety of complex dynamic behavior in pinwheel ASI, while at the same time offering considerable control by choice of clock protocol.

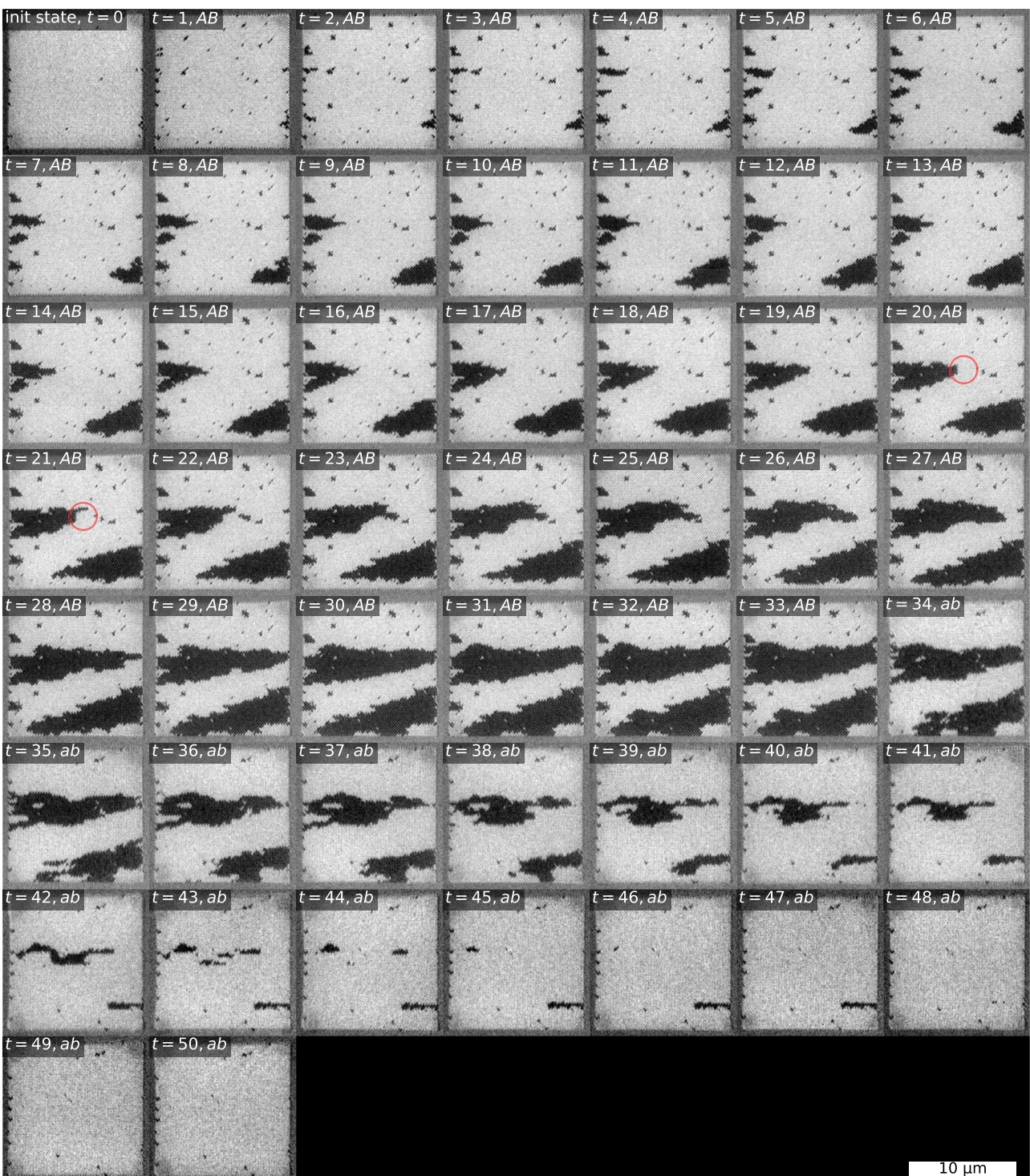

**Fig. 7 | XMCD-PEEM images of all steps from the relevant unipolar clock protocol series.** Time starts at $t = 0$, and is incremented by 1 for each clock step, with clock pulses indicated by the labels. The black (rightwards) domains grow with application of $AB$ clocking, and quickly reverses with $ab$ clocking. Red circle highlights: The short, vertical domain wall terminating the black domain in the center region of snapshot $t = 20$ exemplifies both avalanching domain growth and a stuck domain wall. In snapshot $t = 21$ the top part of the domain wall has progressed in an avalanche to form a finger extension of the domain, while the bottom part of the domain wall remains as before.

## Discussion

We have introduced astroid clocking, a scheme for field-driven evolution in nanomagnetic metamaterials. The method exploits the shape and orientation of the nanomagnet switching astroids, combined with local dipolar coupling, to selectively address subsets of the nanomagnets. Importantly, astroid clocking only requires global fields, yet offers a remarkable degree of control at the microscopic scale. Pulsing specific fields in sequence results in clocked dynamics that are both gradual and discrete in time. Furthermore, considerable control of the dynamics is available through choice of clock protocol.

This work demonstrates how astroid clocking can be used to control the growth and reversal of ferromagnetic domains in pinwheel ASI.

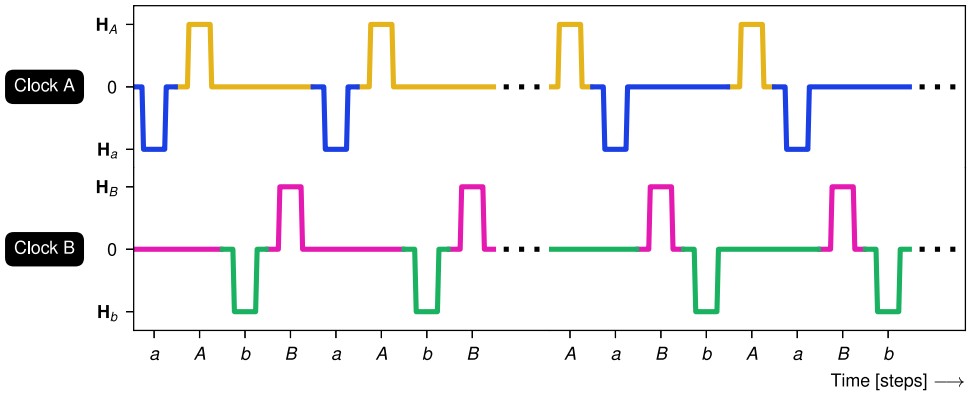

**Fig. 8 | Clock diagram of bipolar *aAbB* clocking followed by its inverse, *AaBb* clocking.** Bipolar clocking employs both positive and negative clock pulses.

**Fig. 9 | Results of growth and reversal with bipolar clocking, and control experiment. a** total magnetization of the ensembles subject to the different bipolar clock protocols. The timeline indicates clock time, labeled by the clock protocol. During the first phase, *AaBb* clocking, the ensembles undergo domain growth and increase in magnetization. The controls, *aA* clocking and *bB* clocking, show no net growth. Further growth (*aAbB* clocking) and reversal (*AaBb* clocking) occur after the controls. **b** magnetic image snapshots of the experimental ensemble, and zoomed in views of the flatspin simulated ensemble, at the specified points in time. The growing domains change morphology during the clock protocol. All XMCD-PEEM images are available in Supplementary Fig. 3. Videos of the experiment and simulation are provided in Supplementary Movies 3 and 4.

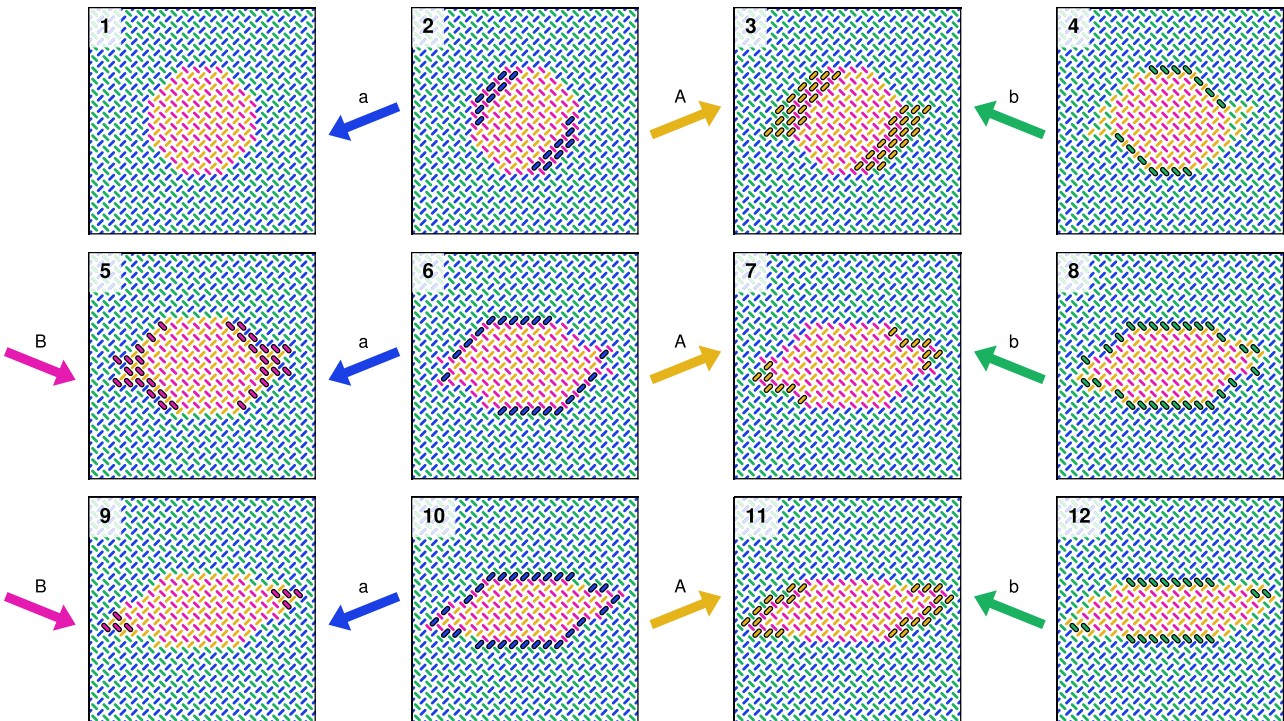

**Fig. 10 | Bipolar *aAbB* clocking of pinwheel ASI.** Each snapshot shows a zoomed-in view of a 50 × 50 system, at different points during a clock protocol. (1) shows the initial state, an orange/pink (rightwards) domain in the center of an otherwise polarized blue/green (leftwards) array. (2–12) show the state during *aAbB* clocking, with simultaneous domain growth (horizontally) and reversal (vertically). As a result the domain gradually changes morphology over time. Magnets that change state between snapshots are highlighted by a solid black outline.

In this system, unipolar clocking results in monotonic domain growth or reversal, while bipolar clocking adds more complex dynamics that include changes to domain morphology. The level of control far exceeds what is possible with conventional ferromagnetic materials, and what has previously been achieved in magnetic metamaterials.

The principles of astroid clocking are not limited to pinwheel ASI, and are pertinent to a range of coupled nanomagnetic systems. Preliminary simulation results of both square and kagome ASI strongly suggest the method is widely applicable. However, further work is needed to establish the most suitable protocols for these geometries. Exploring the clocked dynamics of established and future nanomagnetic metamaterials is an exciting research direction. Notably, with a large variety of possible clock protocols, the method opens for the exploration of new and exotic metamaterial states.

Astroid clocking offers significant control of ASI dynamics in both time and space, enabling the full richness of emergent behavior to be explored and exploited. The method enables entirely new directions in both fundamental and applied metamaterial research, and is key for the development of nanomagnetic technology.

## Methods
### Sample fabrication details
The samples are arrays of permalloy nanomagnets fabricated in pinwheel ASI geometries on a silicon substrate. The resist mixture, 1:2 CSAR 62:anisole, is spin-coated onto the substrate at 4000 rpm, achieving a thickness of ~100 nm. Following coating, samples are soft baked at 150 °C for 1 min. The desired patterns, arrays of 220 nm × 80 nm stadium shaped nanomagnets in 30 × 30 and 50 × 50 pinwheel geometries, with lattice spacing 255 nm and 248 nm, respectively, are then exposed using the Elionix ELS-G100 EBL system. Samples are post-exposure baked at 150 °C for 1 min. The patterned resist is developed using AR600-546 for 1 min, rinsed with isopropanol, and nitrogen dried. Permalloy ($Ni_{0.79}Fe_{0.21}$) is deposited to a thickness of

25 nm via electron beam evaporation using a Pfeiffer Vacuum Classic 500 system, and capped with a 2 nm aluminium layer. Finally, the samples undergo ultrasound-assisted lift-off using a dedicated stripper (AR600-71), leaving behind the patterned permalloy nanomagnets. Post-fabrication, the precision and quality of the fabricated nanomagnet arrays are inspected using scanning electron microscopy (SEM). This SEM inspection confirmed that the permalloy nanomagnets are properly formed, free-standing, and without significant defects.

### XMCD-PEEM and clocking procedure
Experimentally realized clocking of fabricated ASIs is carried out under magnetic microscopy inspection. We use a photoemission electron microscope with x-ray magnetic circular dichroism (XMCD-PEEM) for magnetic contrast to observe single magnet states of the ASI ensembles[35]. An in-plane, bi-axial quadrupole magnet with two pairs of coils and a split 2D-yoke provides astroid clocking fields[36]. The signal at the Fe $L_3$ edge is exploited for ferromagnetic XMCD contrast.

The orientation of the ASI ensembles, applied magnetic fields, and XMCD contrast is carefully selected. Samples are mounted with top and bottom ensemble edges parallel to the synchrotron light, with each nanomagnetic element oriented ±45° to the light. This orientation guarantees balanced magnetic contrast for nanomagnets of both sublattices $L_a$ and $L_b$. The in-plane field direction is given relative to the incoming x-ray illumination, with angle values increasing counter-clockwise. Consequently, the field directions and ensemble orientation align with the illustration in Fig. 1, with an added light axis (providing magnetic contrast) parallel to the $h_x$-axis.

The general experimental procedure is to initialize the ASI system, then apply clock protocols interspersed with magnetic imaging. We initialize the system by applying a strong, polarizing magnetic field

(72 mT along 180°), followed by two smaller fields, (18 mT along 0° and 3.5 mT along 180°) to demagnetize the yoke. For the bipolar clocking, however, the polarizing field strength is 82 mT. The difference in field strength is due to observed differences in the ensemble coercivity. Successful initialization is confirmed by imaging a fully polarized ensemble (fully bright contrast (leftwards), as in snapshot $t = 0$ of Fig. 7) and the absence of remaining image translation in the PEEM (indicating a demagnetized yoke).

After initialization, we perform steps of the clock protocols by alternating the application of clock pulses A, B, a or b. Each *step* of a clock protocol comprises at least one *clock pulse* (ramping the applied field to $\mathbf{H}_i$, holding the max field value, ramping down to zero applied field), and a magnetic contrast image acquisition. The value of $H$ that defines the $\mathbf{H}_i$ magnitudes is 62 mT for the unipolar clocking, and 75 mT for the bipolar clocking. After applying the first cycle of a clock protocol, before imaging, we shift the image, using the electron microscope optics, to re-center the ensemble, compensating for a small remanent magnetization in the yoke. We carry out multiple cycles, each consisting of applying clock pulses and capturing an image, while maintaining the same image shift throughout.

In addition to the growth and reversal protocols, we conduct a control experiment by applying repeated clock pulses of A and B separately.

### flatspin simulations

Numerical simulations were done using flatspin, a large-scale ASI simulator[33]. flatspin approximates each nanomagnet as a point dipole with position $\mathbf{r}_i$ and orientation $\theta_i$. Each dipole then has two possible magnetization directions along $\theta_i$, i.e., a binary macrospin $s_i \in \{-1, +1\}$.

Each spin $i$ is influenced by a total field $\mathbf{h}_i = \mathbf{h}_{dip}^{(i)} + \mathbf{h}_{ext}^{(i)} + \mathbf{h}_{th}^{(i)}$, where $\mathbf{h}_{dip}^{(i)}$ is the total dipolar field from neighboring magnets, $\mathbf{h}_{ext}^{(i)}$ is a global or local external field, and $\mathbf{h}_{th}^{(i)}$ is a stochastic magnetic field representing thermal fluctuations in each magnetic element. The total dipolar field is given by the magnetic dipole-dipole interaction,

$$\mathbf{h}_{dip}^{(i)} = \alpha \sum_{j \neq i} \frac{3\mathbf{r}_{ij}(\mathbf{m}_j \cdot \mathbf{r}_{ij})}{|\mathbf{r}_{ij}|^5} - \frac{\mathbf{m}_j}{|\mathbf{r}_{ij}|^3}, \tag{1}$$

where $\mathbf{r}_{ij} = \mathbf{r}_i - \mathbf{r}_j$ is the distance vector from spin $i$ to $j$, and $\alpha$ scales the dipolar coupling strength between spins. The coupling strength $\alpha$ is given by $\alpha = \frac{\mu_0 M}{4\pi a^3}$, where $a$ is the lattice spacing, $M$ is the net magnetic moment of a single magnet, and $\mu_0$ is the vacuum permeability.

Nanomagnet switching (magnetization reversal) occurs if the total field is directed against the current magnetization $\mathbf{m}_i$ and the magnitude of the field exceeds the coercive field $h_c$. flatspin employs a generalized Stoner-Wohlfarth model, where $h_c$ depends on the angle of the total field $\mathbf{h}_i$ with respect to the magnet orientation. Associated with each magnet is a switching astroid, which describes $h_c$ in terms of the parallel (easy axis) and perpendicular (hard axis) component of the total field, $\mathbf{h}_\parallel$ and $\mathbf{h}_\perp$. The shape of the switching astroid is described by the equation

$$\left(\frac{h_\parallel}{bh_k}\right)^{2/\gamma} + \left(\frac{h_\perp}{ch_k}\right)^{2/\beta} = 1, \tag{2}$$

where $h_k$ denotes the coercive field along the hard axis. The parameters $b$, $c$, $\beta$, and $\gamma$ adjust the shape of the astroid: $b$ and $c$ define the height and width, respectively, while $\beta$ and $\gamma$ adjust the curvature of the astroid at the easy and hard axis, respectively. Astroid parameters are typically tuned to obtain a shape that agrees with results from micromagnetic simulations.

Fabrication imperfections are modelled as variation in the coercive fields $h_k^{(i)}$, which are sampled from a normal distribution $\mathcal{N}(h_k, \sigma)$, where $\sigma = k_{disorder} \cdot h_k$ and $k_{disorder}$ is a user-defined parameter.

Dynamics are modeled using a deterministic single spin flip strategy. At each simulation step, the total magnetic field $\mathbf{h}_i$ is

calculated. Next, we obtain a list of spins that may flip, according to the switching astroid. Finally, the spin which is furthest outside its switching astroid is flipped. The dipolar fields are recalculated after every spin flip, and the above process is repeated until there are no more flippable spins. This relaxation process is performed with constant external and thermal fields.

In this work, a global external field is used ($\mathbf{h}_{ext}^{(i)} = \mathbf{h}_{ext}$), and thermal fluctuations are assumed to be negligible ($\mathbf{h}_{th}^{(i)} = 0$).

The coupling strength $\alpha = 0.0013$ was estimated to match the experimental results from the $50 \times 50$ fabricated pinwheel sample. The value of $\alpha = 0.0013$ is lower than predicted by theory ($\alpha \approx 0.0025$), which is likely due to demagnetizing oxidation of the permalloy. A partially oxidized nanomagnet will have a reduced magnetic moment and a smaller effective size as the surface layer is no longer ferromagnetic. The smaller $30 \times 30$ sample used in Fig. 9 had a slightly larger magnet spacing and $\alpha = 0.0012$ was used in this case.

For the simulation studies, a field strength $H = 76.5$ mT and no disorder was used. Simulations accompanying the experimental results used a slightly lower field strength of $H = 75.8$ mT for Fig. 6 and $H = 75.9$ mT for Fig. 9.

Switching parameters were estimated from micromagnetic simulations of a $220\,nm \times 80\,nm \times 25\,nm$ stadium magnet using mumax[37], namely $h_k = 0.2$ T, $b = 0.38$, $c = 1$, $\beta = 1.3$, and $\gamma = 3.6$. Other parameters include $k_{disorder} = 4\%$ and a neighbor distance of 10.

## Data availability

The XMCD-PEEM data, XAS data, and simulation results generated in this study have been deposited in the Zenodo database under accession code 10044134 at https://zenodo.org/doi/10.5281/zenodo.10044133.

## Code availability

Numerical simulations were performed using the open-source flatspin simulator (https://flatspin.gitlab.io/). Simulation details are included as part of the dataset (see above).

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

## Acknowledgements

These experiments were performed at the CIRCE beamline at ALBA Synchrotron with the collaboration of ALBA staff. This work was funded by the Norwegian Research Council through the IKTPLUSS project SOCRATES (Grant no. 270961) and the TEKNOKONVERGENS project SPrINTER (Grant No. 331821), and by the EU FET-Open RIA project SpinENGINE (Grant No. 861618). The Research Council of Norway is acknowledged for the support to the Norwegian Micro- and Nano-Fabrication Facility, NorFab, project number 295864. Simulations were executed on the NTNU EPIC compute cluster[38]. M.A.N., M.F. and M.W.K. acknowledge funding from MCIN through grant number PID2021-122980OB-C54 and M.W.K. also acknowledges support through Marie Sklodowska-Curie grant agreement No. 754397 (DOC-FAM) from EU Horizon 2020.

## Author contributions

J.H.J. and A.S. conceived and designed the study and contributed equally to this work. J.H.J. did the initial discovery in simulations and did the simulation study. A.S. fabricated the samples. A.S. led the XMCD-PEEM experiments and J.H.J., I.B., A.P., G.T. and E.F. contributed to these measurements. M.F., M.A.N. and M.W.K. provided support during XMCD-PEEM measurements and PEEM vector magnet operation. A.S. and J.H.J. performed the analysis with assistance from I.B. and A.P. E.F. and G.T. oversaw the project and provided feedback and direction throughout. J.H.J. and A.S. wrote the manuscript with input from all authors.

## Funding

## Competing interests

The authors declare no competing interests.
