## [Peer Review File · Nature Communications]

Reviewers' Comments:

Reviewer #1:

Remarks to the Author:

The manuscript offers a very nice and comprehensive simulation and experimental verification of astroid clocking pinwheel ASI arrays. Most noteworthy, this manuscript helps elucidate complications in the demagnetization process in ASI materials and offers a well thought out method for optimizing the demag procedure in the pinwheel ice, but the authors also tease that it may be effective with a variety of lattice geometries. If correct in this statement, the selective flipping of island moments in any ASI system would be a very exciting result for the ASI community, the magnetic nanostructure community, and potentially other fields as well. The manuscript is very well written, the figures are excellent (for the most part, see below) and deserves the visibility nature communications will offer. Therefore, I recommend this paper for publication with very minor comments/corrections (see below).

- The authors mention that astroid clocking could be extended to a variety of lattice geometries, but do not mention if they have tried this. I am curious if this method is so useful for the pinwheel lattice due to its chiral geometry and with more complicated geometries (e.g. kagome and the vertex frustrated systems) it will not work so well. Could the authors offer a little more description about this?
- There seems to be an issue with Figure 4b (and 8b) in the manuscript (this might be a localized problem to just me), it doesn't show the blue data points. They appear in the source file, but this is something that should be checked thoroughly before publication. Also, axes scales for the parallel and perpendicular fields would be a nice addition to Fig 4b (and 8b) to get a better sense of the spread in the data.
- Considering that your domains grow from the edges, have you tried a similar experiment with a larger array size, so as to eliminate any edge effects? Or, add a missing column of islands to try to start the nucleation from a particular area?

Reviewer #2:

Remarks to the Author:

The paper pertains to the characteristic shape of the switching threshold map for ferromagnets - the Stoner-Wohlfarth astroid, and the way the nanofabrication can be used to tailor that map. Once that is established the team have developed innovative field sequences that can achieve a very controlled ratchet-like magnetic reversal. The experimental data is comprehensive and very good quality. The simulations are also excellent. The agreement between the two is as good as could be expected. The reversal is very sensitive to disorder and the experimental sample has unknown (but very specific) quenched disorder whereas in the simulations one can only use an averaged random distribution which will not match perfectly. However, the matching is good enough to see that all other aspects are captured perfectly and allow understanding of the phenomenon. It is a set of data that certainly merits publication in Nature Comms, however I do have some reservations about the manuscript in its current form. I am an expert in the specific field, and I have reasons why I think the data is of a suitable interest for publication in Nature Comms, but I don't think a non-specialist reader would be able to work out why the results are important from the manuscript in its current form.

A more minor point is that while there are good historical reasons why the authors are referring to the threshold map as an astroid, an astroid is a specific geometric shape (a four-pointed shape with curved edges). The name is accurate for 3D bulk or 2D films, but the point of the nanofabrication is that it dramatically distorts the threshold map such that it is no longer an astroid. This problem is compounded by the fact that the total shape in Fig 1b actually looks extremely close to a true astroid, so a non-specialist reader who googles what an astroid is, is likely to think it means the whole thing, whereas the authors intend "astroid" to mean just one half - either the solid lines or the dotted lines. I suggest the authors either clarify that point or modify the nomenclature, but leave that up to them.

In summary, I think the paper could be published in Nature Comms after revisions to make the motivation for and significance of the work more readily understandable to a non-specialist.

Reviewer #3:

Remarks to the Author:

The paper entitled "Clocked dynamics in artificial spin ice" by Johannes H. Jensen, et al., describes a clocking method named "Astroid Clocking". The astroid clocking gives us a stable way to control magnetization of ASI, which is relatively difficult to control with uniaxial field or rotating field. The proposed method will extend the way of using ASI as an information processing device. However, there are some concerns. I will recommend publishing this paper after a minor revision.

According to the 2-fold rotational symmetry of the stadium shape, it is possible to select the switching magnet even if we use the clock angle of 45 degree. However, in this paper, the system only works with clock field angle from 10 degree to 35 degree (page 4, third paragraph). What will happen if we use a clock field with angle of 45 degree?

If the selectivity of the domain wall will lose but a domain still show some breathing motion, then I think this behaviour is still useful. The oscillating motion can be used for a physical reservoir. If simulation is possible, I recommend adding a short description of simulation results with 45 degree.

Or if the restriction of the flat-spin simulator, that the magnetization is binary does not allow to perform a realistic simulation in this region, please mention it.

Minor concerns.

The horizontal and vertical axes in Fig. 4a, Fig. 8a and Fig. S1c are parallel to the easy/hard axes of each nano-magnet, respectively. This is described in the method section (page 18), however it would be easier to understand if it were written in the caption of Fig. 4. It will also be noted that the direction of the magnetization is the same as the positive direction of the parallel axis.

The definition of nearest neighbor degree in Fig. S1 is unclear. Is the number of the nearest neighbor degree of Fig. S2 is two? It will helpful that the number is written in the caption of Fig. S2.

I think the Fig. S2 is very important for understanding the magnetic domain wall migration. For easier understanding, it would be helpful if the HA is shown in the figure.

The plot point of Fig. 4b and Fig.8b is missing in the merged PDF. The individual figure file shows the correct plot points. This may be a bug in the upload system...

Feedback from Reviewer #1:

The manuscript offers a very nice and comprehensive simulation and experimental verification of astroid clocking pinwheel ASI arrays. Most noteworthy, this manuscript helps elucidate complications in the demagnetization process in ASI materials and offers a well thought out method for optimizing the demag procedure in the pinwheel ice, but the authors also tease that it may be effective with a variety of lattice geometries. If correct in this statement, the selective flipping of island moments in any ASI system would be a very exciting result for the ASI community, the magnetic nanostructure community, and potentially other fields as well. The manuscript is very well written, the figures are excellent (for the most part, see below) and deserves the visibility nature communications will offer. Therefore, I recommend this paper for publication with very minor comments/corrections (see below).

We thank the Reviewer for the positive feedback and recommendation for publication.

The authors mention that astroid clocking could be extended to a variety of lattice geometries, but do not mention if they have tried this. I am curious if this method is so useful for the pinwheel lattice due to its chiral geometry and with more complicated geometries (e.g. kagome and the vertex frustrated systems) it will not work so well. Could the authors offer a little more description about this?

We have added a sentence to the final paragraph of the Introduction, where we briefly mention preliminary results of astroid clocking of the square and kagome geometries. We also added two sentences to the penultimate paragraph of the conclusion, where we make a similar remark and note that further work is needed for these geometries. However, we need to limit the scope of this paper, and while the other geometries showed promising results, they did not yield any further insight into the clocking mechanism. The other geometries are also more sensitive to disorder, which made pinwheel the best alternative.

There seems to be an issue with Figure 4b (and 8b) in the manuscript (this might be a localized problem to just me), it doesn't show the blue data points. They appear in the source file, but this is something that should be checked thoroughly before publication.

We have been able to reproduce the problem by opening the PDF in Google Chrome. The issue seems to be related to the large number of graphical elements in the figure. We have solved the problem in Google Chrome by optimizing the figure, but urge the Editor to double check on their part.

Also, axes scales for the parallel and perpendicular fields would be a nice addition to Fig 4b (and 8b) to get a better sense of the spread in the data.

This was a good suggestion, and we have added axes scales to Fig 4b and 8b.

Considering that your domains grow from the edges, have you tried a similar experiment with a larger array size, so as to eliminate any edge effects? Or, add a missing column of islands to try to start the nucleation from a particular area?

Yes, we have indeed investigated means of controlling the nucleation of domains. We have added a sentence at the end of paragraph 3, section 5, which highlights some methods to achieve spatial control of nucleation.

Feedback from Reviewer #2:

The paper pertains to the characteristic shape of the switching threshold map for ferromagnets - the Stoner-Wohlfarth astroid, and the way the nanofabrication can be used to tailor that map. Once that is established the team have developed innovative field sequences that can achieve a very controlled ratchet-like magnetic reversal. The experimental data is comprehensive and very good quality. The simulations are also excellent. The agreement between the two is as good as could be expected. The reversal is very sensitive to disorder and the experimental sample has unknown (but very specific) quenched disorder whereas in the simulations one can only use an averaged random distribution which will not match perfectly.

However, the matching is good enough to see that all other aspects are captured perfectly and allow understanding of the phenomenon.

Thank you, we are very grateful for the positive comments on our work.

It is a set of data that certainly merits publication in Nature Comms, however I do have some reservations about the manuscript in its current form. I am an expert in the specific field, and I have reasons why I think the data is of a suitable interest for publication in Nature Comms, but I don't think a non-specialist reader would be able to work out why the results are important from the manuscript in its current form.

We have made significant changes to the Abstract and Introduction to make the impact of the work more accessible to a non-specialist. We have also revised the Conclusion to better highlight the potential of the work.

A more minor point is that while there are good historical reasons why the authors are referring to the threshold map as an astroid, an astroid is a specific geometric shape (a fourpointed shape with curved edges). The name is accurate for 3D bulk or 2D films, but the point of the nanofabrication is that it dramatically distorts the threshold map such that it is no longer an astroid. This problem is compounded by the fact that the total shape in Fig 1b actually looks extremely close to a true astroid, so a non-specialist reader who googles what an astroid is, is likely to think it means the whole thing, whereas the authors intend "astroid" to mean just one half – either the solid lines or the dotted lines. I suggest the authors either clarify that point or modify the nomenclature, but leave that up to them.

The Reviewer raises an excellent point regarding the potential confusion when using the term astroid to describe distorted astroid shapes. We have decided to keep the astroid nomenclature for the imperfect/distorted astroid shapes. To avoid confusion, we have added two sentences to the third paragraph of section 2, where we clarify this explicitly. We have also added clarifying words to the caption of Fig. 1 to highlight that there are two switching astroids.

Feedback from Reviewer #3:

The paper entitled "Clocked dynamics in artificial spin ice" by Johannes H. Jensen, et al., describes a clocking method named "Astroid Clocking". The astroid clocking gives us a stable way to control magnetization of ASI, which is relatively difficult to control with uniaxial field or rotating field. The proposed method will extend the way of using ASI as an information processing device. However, there are some concerns. I will recommend publishing this paper after a minor revision.

We thank the Reviewer for the positive recommendation.

According to the 2-fold rotational symmetry of the stadium shape, it is possible to select the switching magnet even if we use the clock angle of 45 degree. However, in this paper, the system only works with clock field angle from 10 degree to 35 degree (page 4, third paragraph). What will happen if we use a clock field with angle of 45 degree? If the selectivity of the domain wall will lose but a domain still show some breathing motion, then I think this behaviour is still useful. The oscillating motion can be used for a physical reservoir. If simulation is possible, I recommend adding a short description of simulation results with 45 degree. Or if the restriction of the flat-spin simulator, that the magnetization is binary does not allow to perform a realistic simulation in this region, please mention it.

The Reviewer points out an insightful question. When the clock field is at 45°, the mechanism breaks down and there is avalanche switching. The Reviewer is correct in that we lose specific domain wall selectivity, but unfortunately this does not result in the breathing motion mentioned. Instead, the system tends to completely polarize during a single clock cycle, mediated by a series of avalanches within each sublattice. We have added a sentence highlighting the failure of these angles at the end of the penultimate paragraph of section 2. Furthermore, we have added three sentences to the fourth paragraph of section 4, explaining the failure mechanism in more detail.

The horizontal and vertical axes in Fig. 4a, Fig. 8a and Fig. S1c are parallel to the easy/hard axes of each nano-magnet, respectively. This is described in the method section (page 18), however it would be easier to understand if it were written in the caption of Fig. 4. It will also be noted that the direction of the magnetization is the same as the positive direction of the parallel axis.

We have added an explicit statement of the relative axes in the captions of Fig. 4, Fig. 8, and Fig. S1, and also added comments on the parallel axis positive direction.

The definition of nearest neighbor degree in Fig. S1 is unclear. Is the number of the nearest neighbor degree of Fig. S2 is two? It will helpful that the number is written in the caption of Fig. S2.

This has been clarified in the revised manuscript. We have explicitly defined the neighbor degree N in the second paragraph of section S1, as well as in the captions of Fig. S1 and Fig. S2.

I think the Fig. S2 is very important for understanding the magnetic domain wall migration. For easier understanding, it would be helpful if the HA is shown in the figure.

We we have added a mention of HA in the caption of Fig. S2. Due to space limitations, we decided not to add HA to the figure itself.

The plot point of Fig. 4b and Fig.8b is missing in the merged PDF. The individual figure file shows the correct plot points. This may be a bug in the upload system...

We believe we have fixed this issue, which seemed to be a problem relating to certain PDF-viewers and a maximum number of vector graphics elements. However, we would again ask the Editor to double check these figures explicitly.

Other changes

In addition to the changes prompted by the reviewers, the manuscript includes a few minor changes: •

Added Data availability section after Methods, referring to a Zenodo archive containing experimental and simulation data.

- Resized figures to conform to Nature's standard figure sizes.